# Leveraging GPT-4 for Automatic Translation Post-Editing

**Vikas Raunak**     **Amr Sharaf**     **Yiren Wang**     **Hany Hassan Awadalla**     **Arul Menezes**

Microsoft Azure AI
Redmond, Washington
{viraunak,amrsharaf,wangyiren,hanyh,arulm}@microsoft.com

## Abstract

While Neural Machine Translation (NMT) represents the leading approach to Machine Translation (MT), the outputs of NMT models still require translation post-editing to rectify errors and enhance quality under critical settings. In this work, we formalize the task of direct translation post-editing with Large Language Models (LLMs) and explore the use of GPT-4 to automatically post-edit NMT outputs across several language pairs. Our results demonstrate that GPT-4 is adept at translation post-editing, producing meaningful and trustworthy edits to translations that help improve its general quality as well as remove different classes of major errors in translations. In particular, human evaluations on assessing edit trustworthiness show that GPT-4 exhibits a large improvement over the prior state-of-the-art LLM. Notably, we improve upon state-of-the-art performance on WMT-22 English-Chinese, English-German, Chinese-English and German-English language pairs using GPT-4 based post-editing, as evaluated by state-of-the-art MT quality metrics. However, we also show that GPT-4 could produce hallucinated edits, thereby urging caution in its use as an expert translation post-editor.

## 1 Introduction

State-of-the-art Neural Machine Translation (NMT) models, trained on web-mined parallel corpora suffer from reliability problems even for higher resource language pairs, despite high average case performance (He et al., 2020; Gupta et al., 2020; Sun et al., 2020; Wang et al., 2021; He et al., 2021; Raunak et al., 2022; Raunak and Menezes, 2022). A number of prior works have demonstrated that the parallel data and model training artifacts in NMT could manifest in terms of catastrophic outputs in rare cases, and the detection of such egregious model behaviors remains a challenging task (Raunak et al., 2021; Tang et al., 2022; Guerreiro et al., 2023b; Xu

et al., 2023; Guerreiro et al., 2023a). Thereby, post-editing neural machine translations remains an important exercise for their use in critical settings across the translation and localization industry. As such, a relevant question to ask is whether Large Language Models (LLMs) such as GPT-3, GPT-4 and PaLM, PaLM2 (Brown et al., 2020; OpenAI, 2023; Anil et al., 2023; Chowdhery et al., 2022), which have demonstrated a wide-range of general purpose reasoning and knowledge-based capabilities could be leveraged for the task of translation post-editing. LLM based automatic translation post-editing could aid in both detecting and fixing translation errors to ensure greater reliability of NMT outputs. Besides alleviating reliability problems in NMT, leveraging LLMs for post-editing could be opportune owing to the advanced multi-lingual understanding capabilities of latest LLMs, in particular GPT-4 (OpenAI, 2023) and potentially, their ability to apply knowledge-based or culture-specific customization to translations (Baker, 2009), which are hard to ascertain only through the source text.

In this work, we explore the efficacy of state-of-the-art LLMs such as GPT-4 on the task of translation post-editing in a *direct* setting, i.e., without any quality-estimation or error detection step applied to the translations prior to post-editing. Our contributions are as follows: (i) We formalize the task of direct translation post-editing with LLMs and posit a set of research questions and measurements to quantify their utility for the goal of improving translations obtained from NMT models. (ii) We demonstrate that GPT-4 produces meaningful, human-judgement aligned edits to state-of-the-art translations with a high degree of trustworthiness. (iii) We demonstrate gains on translation quality over best systems from WMT-22 across a number of language pairs on the WMT-22 benchmark (Koehn et al., 2022), improving upon the state-of-the-art translation performance on English-

| **Example of Translation Post-Editing using GPT-4 (Verbatim Outputs under the CoT Setting)** |
|:---|

**Source**: They were addressed to her son, who has autism and lives in a private care facility, she said. But instead of her son's name inside when you opened them, the letters said Dear Maine's Department of Health and Human Services – in Cincinnati, she told local media.

**Translation**: Sie waren an ihren Sohn gerichtet, der Autismus hat und in einer privaten Pflegeeinrichtung lebt, sagte sie. Aber anstelle des Namens ihres Sohnes im Inneren, als Sie sie öffneten, hieß es in den Briefen Dear Maine Dear Maine 's Department of Health and Human ServicesServices - in Cincinnati, sagte sie den lokalen Medien.

---

**Proposed Improvements**:

1. Remove the repetition of "Dear Maine" in the German translation.
2. Correct the translation of "Dear Maine's Department of Health and Human Services" to "Sehr geehrtes Department of Health and Human Services von Maine".
3. Replace "ServicesServices" with "Services".
4. Add a comma after "Cincinnati" for better sentence structure.

**Improved Translation**:

Sie waren an ihren Sohn gerichtet, der Autismus hat und in einer privaten Pflegeeinrichtung lebt, sagte sie. Aber anstelle des Namens ihres Sohnes im Inneren, als Sie sie öffneten, hieß es in den Briefen "Sehr geehrtes Department of Health and Human Services von Maine" - in Cincinnati, sagte sie den lokalen Medien.

Table 1: **Post-Editing Task Example**: GPT-4 produces edits that result in improving the general quality of the translation as well as in removing undesirable artifacts across a range of NMT systems, as quantified in Section 3.

Chinese, English-German, Chinese-English and German-English using GPT-4 based post-editing, as evaluated by state-of-the-art MT quality metrics.

## 2 The Translation Post-Editing Task

**Task Definition:** We formalize the post-editing task in a generative setting as follows: given a Source ($S$) and a Translation ($T$), propose improvements over the translation ($E$) and generate the translation with the proposed edits ($T'$), i.e.:

$$(S, T) \rightarrow E + T'$$

Under this task setting, $E$ represents the improvements or the edits that are verbalized by a LLM. In the absence of $E$, the task is reduced to simply generating the improved translation without any intermediate reasoning chain or *Chain of Thought* (CoT) (Wei et al., 2022b; Kojima et al., 2022). Throughout this work, we refer to the post-editing task in the above *zero-shot* CoT setting as post-editing with CoT and the setting without $E$ as post-editing without CoT. Table 1 shows an input-output example for the post-editing task under the CoT setting. Additionally, throughout this work, we refer to $Z$ as the zero-shot translation of a given source ($S$) obtained from the same LLM that is employed for the post-editing task. Through this formalization, we posit and investigate the following research questions (RQ):

**(RQ1) Nature of Post-Edited Translations:** LLMs have been shown to generate high quality, state-of-the-art translations (Hendy et al., 2023) across a number of language pairs in a zero-shot setting. As such, during post-editing, LLMs could generate a translation that is incognizant of the provided initial translation. Hence, we investigate whether during translation post-editing LLMs generate the improved translations from scratch (i.e. only based on the source $S$) or do they edit the initial translation $T$ provided as per the instructions of the task. A related question is characterizing the role of CoT in determining the nature of the post-edited translation, i.e. whether the post-edited translation is closer to the initial translation or to the zero-shot translation produced by the LLM.

**(RQ2) General Quality Improvements:** Do the post-edited translations produced by LLMs lead to general quality improvements as measured by state-of-the-art MT quality metrics? Another related question is whether the post-editing chain-of-thought is helpful towards translation quality improvement? Even though zero-shot chain-of-thought has been demonstrated to be effective across reasoning tasks, we hypothesize that trans-

| Research Question | Measurement | Datasets |
|---|---|---|
| Nature of the Post-Edited Translation | TER($T'$, Zero-Shot) vs TER($T'$, $T$) | WMT-22 |
| General Quality Improvements | COMET*($S$, $T$) vs COMET*($S$, $T'$) | WMT-22 |
| Edits On Human Annotated Error Spans | Edit Efficacy over Error Spans (E3S) | WMT-20, 21, 22 |
| Trustworthiness of the Proposed Edits | Edit Realization Rate (ERR) | WMT-22 |

Table 2: **Measuring Post-Editing Efficacy**: Given, the Source $S$, Translation $T$ and the Post-Edited Output $T'$, we explore the four research questions in section 2 through experiments on the corresponding datasets, using the proposed measurements described in detail in Section 4. COMET* represents any of the COMET MT metrics.

lation post-editing task might not require the same degree of variable computation that makes CoT effective (Kojima et al., 2022), owing to the lack of multiple reasoning steps involved in the task.

**(RQ3) Modifying Human Annotated Error Spans:** Are LLMs capable of modifying human annotated translation error spans during the post-editing step? A high frequency of modifications made to the human annotated error spans, especially if it is accompanied by general quality improvements, would signify a greater correlation with human judgement in identifying errors in translations.

**(RQ4) Trustworthiness of the Proposed Edits:** Do the edits proposed as CoT actually appear in the improved translation produced by LLMs, in the post-editing with CoT setting? It is quite conceivable that LLMs might make edit proposals or produce chain of thought that is not *realized* in the final post-edited translation produced by the same model (Ye and Durrett, 2022; Turpin et al., 2023). However, if the post-editing explanation or CoT is a desiderata of the translation post-editing process, it becomes critical to examine the fidelity of the proposed edits in addition to the final translation quality. A higher realization rate of the proposed edits would also help establish the trustworthiness of the LLM as an expert translation post-editor.

In the next sections, we explore the above four research questions using the state-of-the-art LLMs. We describe our experimental settings in Section 3 & present the results in Section 4.

## 3 Experimental Settings

**Datasets:** We experiment with WMT-22 General MT translation task datasets (Kocmi et al., 2022) as well as with WMT-20 and WMT-21 News translation task submissions annotated with MQM (Mul-

tidimensional Quality Metrics Framework) errors[1] Freitag et al. (2021). For the post-editing experiments pertaining to the MQM annotated WMT-20 and WMT-21 system outputs, we experiment with samples that have a major error as an annotation, whereas we experiment with the full WMT-22 datasets throughout. We use the latest WMT-22 test sets for the majority of our experiments, the curation of which falls beyond the training cut-off dates for GPT-4 and other LLMs under investigation[2].

**LLMs and Baselines:** We experiment with GPT-4 and gpt-3.5-turbo in our experiments. These models represent the most capable publically available LLMs (Liang et al., 2022). We use a prompt that describes the system role as a translation post-editor and under the CoT setting, instruct the LLM to propose improvements to the provided translation ($T$) of a given source ($S$), before producing the final post-edited translation ($T'$). For post-editing, we experiment under three settings: (i) post-editing with CoT, (ii) post-editing without CoT and (iii) post-editing with Structured-CoT (SCoT). The SCoT baseline uses the MQM annotation instructions from Freitag et al. (2021) to produce the intermediate CoT in the form of an MQM annotation over the source-translation pair. We describe the prompts used for the three baselines in appendix C. For producing the initial translations on WMT-22, we use Microsoft-Translator, one of the strongest publically available MT systems (Raunak et al., 2022; Hendy et al., 2023). For WMT-20 and WMT-21 systems, we take the translations provided by the different NMT systems and annotated in Freitag et al. (2021) as the initial translations upon which post-editing is applied.

---

[1] https://github.com/google/wmt-mqm-human-evaluation

[2] LLMs: https://platform.openai.com/docs/models

**Metrics and Evaluation:** For each of the four research questions posed, we use the metrics highlighted in Table 2. We explain these measurements in the relevant sections. For general quality measurements, we use four COMET (Rei et al., 2020) models[3]: the reference-free COMET-QE (*wmt20-comet-qe-da*), COMET-KIWI (*wmt-22-cometkiwi-da*) Quality Estimation (QE) models and the reference-based COMET-20 (*wmt20-comet-da*) and COMET-22 (*wmt22-comet-da*) models. To measure the similarity of translations, we use the Translation Edit Rate (TER) (Snover et al., 2006) implementation from SacreBLEU (Post, 2018).

## 4 Results and Measurements

### 4.1 Nature of Post-Edited Translations

To measure whether the post-edited translations produced by LLMs adhere to editing the initial translations provided, we compute the Translation Edit Rate (TER) (Snover et al., 2006) of the post-edited translation against the zero-shot translations obtained using the same LLM, and compare it with the TER of the post-edited translation against the initial translation. A higher value of TER ($T'$, $T$) implies that the post-edited translation ($T'$) is closer to the initial translation ($T$) and that the LLM adheres to the task of editing the initial translation.

| PE Setting | TER ($T'$, $Z$) | TER ($T'$, $T$) |
|---|---|---|
| With CoT | 92.0 | **70.3** |
| Without CoT | **84.6** | 94.5 |

Table 3: **WMT-22 En-Zh**: The post-edited translations ($T'$) are closer to the initial translations ($T$) than the zero-shot translations ($Z$) in the CoT setting.

| PE Setting | TER ($T'$, $Z$) | TER ($T'$, $T$) |
|---|---|---|
| With CoT | 42.9 | **22.0** |
| Without CoT | **38.1** | 34.9 |

Table 4: **WMT-22 Zh-En**: The post-edited translations ($T'$) are closer to the initial translations ($T$) than the zero-shot translations ($Z$) in the CoT setting.

**Impact of CoT:** Table 3 describes our results on WMT-22 En-Zh and Table 4 describes our results on Zh-En with post-editing using GPT-4. We find that CoT constrains the final translations to be closer to the initial translation. In the post-editing

---

[3]COMET: https://github.com/Unbabel/COMET

setting without CoT, the final translation is closer to the zero-shot translation, even though the TER difference is much smaller than the difference in the CoT setting.

**Discussion** We find that the above results hold true across different metrics such as edit distance, BLEU (Post, 2018) or ChrF (Popović, 2015) as well as across WMT-22 language pairs and gpt-3.5-turbo. This result also shows a peculiar side-effect of the post-editing task under the CoT setting – that post-editing a system translation might end up leading to a lower quality final translation if the initial translation is lower in quality than the zero-shot translation quality of the LLM under consideration. In the next sub-section, we evaluate GPT-4 under different post-editing settings in terms of general quality improvements.

### 4.2 General Quality Improvements

We compare the translation quality of the post-edited translation against the initial translation using both reference-free and reference-based state-of-the-art neural MT quality metrics.

**Results:** Tables 5, 7, 6 and 8 provide the results of the experiments done on the WMT-22 test sets. Throughout, we find that post-editing under both CoT and direct settings leads to improvements over high-quality initial translations obtained through MS-Translator. Further, Tables 5, 7, 6 and 8 show that direct post-editing of MS-Translator outputs with GPT-4 consistently improves upon the WMT-22-Best translation system quality. We also find that gpt-3.5-turbo consistently underperforms GPT-4 as well as the quality of initial translations, demonstrating a qualitative jump in post-editing efficacy of GPT-4.

### 4.3 Edits On Human Annotated Error Spans

We use the MQM annotated WMT-22 system outputs provided by Freitag et al. (2021) and measure whether the post-edited translation modifies the translation error span as annotated by human annotators. For each of the Major MQM error spans modified, we record a score of 1, else a score of 0. The final score reported, named Edit Efficacy over Erroneous Error Spans (E3S) is higher if more of the erroneous spans have been modified in the post-edited translation. The E3S metric is reported as a percentage over the test set.

| System | COMET-KIWI | COMET-QE | COMET-22 | COMET-20 |
|---|---|---|---|---|
| WMT-Best | 81.38 | 39.96 | 85.04 | 56.60 |
| MS Translator | 81.04 | 38.64 | 84.68 | 55.28 |
| **MS Translator + GPT-4** | **81.66** | **42.15** | **85.41** | **58.21** |
| MS Translator + GPT-4-CoT | 81.40 | 41.05 | 85.28 | 57.84 |
| MS Translator + GPT-4-SCoT | 81.39 | 41.40 | 85.18 | 57.45 |
| MS Translator + GPT-3.5-CoT | 79.32 | 41.56 | 82.71 | 44.82 |
| GPT-4-Zero-Shot | 81.51 | 41.36 | 85.26 | 57.53 |

Table 5: **General Quality Improvements on WMT-22 De-En:** The + sign reflects that the post-editing is applied on the initial translations produced by the given System. MS-Translator + GPT-4 obtains the best performance.

| System | COMET-KIWI | COMET-QE | COMET-22 | COMET-20 |
|---|---|---|---|---|
| WMT-Best | 77.66 | 23.98 | 81.02 | 45.21 |
| MS Translator | 77.58 | 23.97 | 80.35 | 40.40 |
| **MS Translator + GPT-4** | **79.75** | **31.84** | **82.79** | **53.42** |
| MS Translator + GPT-4-CoT | 79.02 | 28.96 | 82.20 | 50.77 |
| MS Translator + GPT-4-SCoT | 78.94 | 28.80 | 82.09 | 49.65 |
| MS Translator + GPT-3.5-CoT | 79.32 | 41.56 | 82.71 | 44.82 |
| GPT-4-Zero-Shot | 79.29 | 30.13 | 82.49 | 51.78 |

Table 6: **General Quality Improvements on WMT-22 Zh-En:** The + sign reflects that the post-editing is applied on the initial translations produced by the given System. MS-Translator + GPT-4 obtains the best performance.

| System | COMET-KIWI | COMET-QE | COMET-22 | COMET-20 |
|---|---|---|---|---|
| WMT-Best | 83.56 | 44.67 | 87.21 | 62.35 |
| MS Translator | 83.35 | 43.48 | 86.78 | 62.06 |
| **MS Translator + GPT-4** | **83.69** | **44.50** | **87.37** | **62.85** |
| MS Translator + GPT-4-CoT | 83.32 | 43.96 | 87.13 | 62.62 |
| MS Translator + GPT-4-SCoT | 83.12 | 44.17 | 86.90 | 61.94 |
| MS Translator + GPT-3.5-CoT | 81.36 | 43.12 | 84.55 | 50.52 |
| GPT-4-Zero-Shot | 82.95 | 44.69 | 86.80 | 60.85 |

Table 7: **General Quality Improvements on WMT-22 En-De:** The + sign reflects that the post-editing is applied on the initial translations produced by the given System. MS-Translator + GPT-4 obtains the best performance.

| System | COMET-KIWI | COMET-QE | COMET-22 | COMET-20 |
|---|---|---|---|---|
| WMT-Best | 82.04 | 32.11 | 86.69 | 61.04 |
| MS Translator | 81.39 | 31.46 | 86.11 | 59.43 |
| **MS Translator + GPT-4** | **82.68** | **34.47** | **87.53** | **63.21** |
| MS Translator + GPT-4-CoT | 81.60 | 32.01 | 86.43 | 59.97 |
| MS Translator + GPT-4-SCoT | 81.81 | 32.56 | 86.56 | 60.20 |
| MS Translator + GPT-3.5-CoT | 79.32 | 41.56 | 82.71 | 44.82 |
| GPT-4-Zero-Shot | 81.73 | 32.61 | 86.51 | 58.66 |

Table 8: **General Quality Improvements on WMT-22 En-Zh:** The + sign reflects that the post-editing is applied on the initial translations produced by the given System. MS-Translator + GPT-4 obtains the best performance.

| System | Initial-QE | PE-QE | E3S |
|---|---|---|---|
| PROMT | 74.58 | **78.90** | 57.86 % |
| M2M100 | 73.06 | **79.64** | 70.30 % |
| QUARTZ | 79.36 | **80.38** | 63.63 % |
| JDExplore | 79.43 | **79.61** | 52.32 % |
| Average | 76.61 | **79.63** | 61.03 % |

Table 9: On **WMT-22 En-De System Outputs** with Major MQM-annotated Errors, Post-Editing with GPT-4 increases translation quality considerably and modifies more than sixty percent of the erroneous spans. The results are agnostic to the MT quality estimation metric.

| System | Initial-QE | PE-QE | E3S |
|---|---|---|---|
| AISP-SJTU | 71.87 | **75.66** | 71.62 % |
| Lan-Bridge | 75.52 | **75.82** | 64.14 % |
| LanguageX | 72.80 | **75.80** | 69.34 % |
| M2M100 | 68.24 | **76.49** | 80.72 % |
| Average | 72.11 | **75.94** | 71.46 % |

Table 10: On **WMT-22 Zh-En System Outputs** with Major MQM-annotated Errors, Post-Editing with GPT-4 increases translation quality considerably and modifies more than seventy percent of the erroneous spans. The results are agnostic to the MT quality estimation metric.

**Results:** Tables 9, 10 and 11 report the results obtained on 14 different WMT-22 NMT system outputs from WMT-22, over three language pairs: English-German, Chinese-English and English-Russian. We find that GPT-4 produces E3S rates above fifty percent with considerably large gains in general quality (measured through COMET-KIWI), signifying that it is able to remove the undesirable artifacts (spans) present in the translations. We repeat this experiment on WMT-20 and WMT-21 MQM annotated System outputs as well in appendix A, with similar results.

## 4.4 Trustworthiness of the Proposed Edits

In a practical setting, the edits ($E$) produced in the post-editing task might be useful to illustrate the changes made by the LLM in the post-edited translation. Therefore, the fidelity of the proposed edits is important for imparting more trust in the LLM based post-editing process. Thereby, the question whether the proposed edits are present in the final improved translation or are hallucinated by the model is of significant practical interest. We quantify this property using Edit Realization Rate (ERR), which measures: of the proposed edits ($E$)

| System | Initial-QE | PE-QE | E3S |
|---|---|---|---|
| eTranslation | 73.52 | **78.61** | 51.42 % |
| HuaweiTSC | 75.30 | **79.22** | 53.35 % |
| M2M100 | 74.80 | **79.93** | 55.70 % |
| PROMT | 75.82 | **78.96** | 48.82 % |
| QUARTZ | 77.87 | **81.02** | 67.81 % |
| JDExplore | **78.91** | 78.45 | 41.18 % |
| Average | 76.04 | **79.37** | 53.04 % |

Table 11: On **WMT-22 En-Ru System Outputs** with Major MQM-annotated Errors, Post-Editing with GPT-4 increases translation quality considerably and modifies more than fifty percent of the erroneous spans, on average. The results are agnostic to the MT QE metrics.

by the LLM in the CoT post-editing setting, how many of the edits were actually *realized* in the improved translation? Since, we do not have any ground truth data to quantify this, we use human evaluation for measuring this property.

**ERR Human Evaluation Protocol:** We ask human annotators (bilingual and native in the target language) to label 100 post-editing samples for both En-De and De-En from the WMT-22 test sets, generated by both gpt-3.5-turbo and GPT-4. The annotator is asked to identify if all of the proposed edits were realized in the final translation (**ALL**) or whether a partial number of proposed edits were realized (**PARTIAL**) or whether none of the proposed edits were realized (**NONE**). The human annotator thereby labels each post-editing sample $(S, T, E, T')$ with one of the three labels.

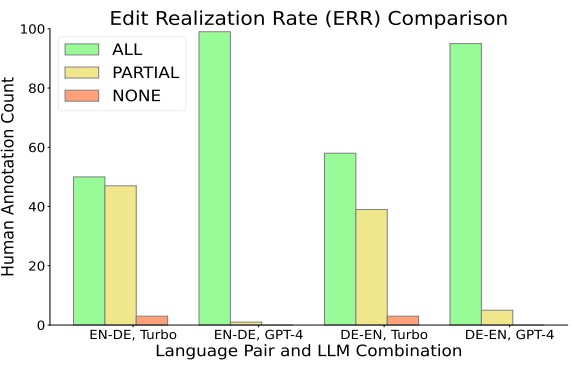

Figure 1: **Edit Realization Rate (ERR) Human Evaluation on WMT-22 En-De and De-En**: GPT-4 obtains higher ERR than gpt-3.5-turbo, imparting greater trustworthiness to the post-editing process under the CoT setting. In conjunction with this result, we also observe that GPT-4 obtains better ERR score distributions on En-Zh translation post-editing, vs Zh-En post-editing.

| **Illustration: Edit Realization Rate Difference between GPT-Turbo and GPT-4** |
|---|
| **Source**: Stephanie Lay said she found the piles of letters from the insurance company stashed in her mailbox between Thursday and Monday.
**Translation**: Stephanie Lay sagte, sie habe die Briefstapel der Versicherung zwischen Donnerstag und Montag in ihrem Briefkasten versteckt gefunden. |
| gpt-3.5-turbo |
| **Proposed Improvements**:
1. The word "stashed" is not adequately translated in the German text.
2. The word "mailbox" is not translated correctly in the German text.
3. The word "piles" is not translated correctly in the German text.
4. The word "found" is not translated adequately in the German text.
5. The word "between" is not translated correctly in the German text.
**Improved Translation**:
Stephanie Lay sagte, sie habe zwischen Donnerstag und Montag im Briefkasten Stapel von Briefen der Versicherung gefunden. |
| GPT-4 |
| **Proposed Improvements**:
1. Replace "Briefstapel der Versicherung" with "Stapel von Briefen der Versicherungsgesellschaft" to better convey the meaning of "piles of letters from the insurance company".
2. Remove "versteckt" as it is not supported in the input and is not necessary for the meaning of the sentence.
**Improved Translation**:
Stephanie Lay sagte, sie habe die Stapel von Briefen der Versicherungsgesellschaft zwischen Donnerstag und Montag in ihrem Briefkasten gefunden. |

Table 12: **Quantifying Edit Realization Rate (ERR)**: The example shows an instance of the Proposed Edits ($E$) and Improved Translation ($T'$) obtained using gpt-3.5-turbo and GPT-4. We find that GPT-4's edit proposals are included in the final translation with a far greater frequency. We quantify this property in Section 4.4.

**ERR Human Evaluation Results:** The results of human evaluations are presented in Figure 1. In general, for both En-De and De-En, there exists a large gap between the ERR distribution of gpt-3.5-turbo and GPT-4. We present a typical post-editing example illustrating this difference in Table 12. Our findings suggest that GPT-4 produces more trustworthy generations for the post-editing task and combined with results in sub-sections 4.1, 4.2 and 4.3, this suggests that GPT-4 could aid in automatic post-editing with considerably greater interpretability. We observed similarly high ERR for En-Zh as well, although the human evaluations still report cases where the edits by GPT-4 are not realized fully, especially in language pairs where the source language is not English (Zh-En and De-En). We present examples of such hallucinated edit proposals in appendix D, showing that GPT-4 might present similar reliability challenges as a post-editor as NMT does in translation.

## 5 Further Discussion

**Post-Editing Across Language Pairs:** We report the GPT-4 post-editing performance under the CoT setting with MS-Translator as the initial translation for several other language pairs in appendix B. In general, the results show that GPT-4 based post-editing leads to consistent gains in translation quality across language pairs, with larger gains for X-E translations.

**Utility of the Chain-of-Thought:** Our results show that the inclusion of the edit proposals (CoT) in the post-editing step is detrimental towards the quality of the post-edited translations, but is useful in constraining the post-edited outputs to the initial translations. Therefore, the necessity of variable computation leveraged by the zero-shot chain-of-thought step is questionable for the post-editing task, even though the edit artifacts produced by the GPT-4 might themselves be valuable for making

the automatic post-editing task more trustworthy. Further, we also found that imposing a specialized structure on the edit proposals in the form of MQM error categories was not valuable towards improving the final post-edited translation quality. On the other hand, we also demonstrated that imposing structure upon the proposed edits in the form of MQM annotations doesn't hurt the general quality of the post-edited translations, suggesting that it might be possible to combine MQM based automatic quality assessment along with post-editing through GPT-4. However, evaluating whether post-editing could be done in conjunction with MQM based quality assessment is beyond the scope of our work and we leave such a joint evaluation to future work.

**Trustworthiness of the Proposed Edits:** We demonstrated that the edits proposed by GPT-4 are realized in the final translation with considerably higher frequency than gpt-3.5-turbo. This shows a quantitative jump in the trustworthiness of the edits proposed by the LLM for the direct automatic post-editing task under our formalization. This jump represents an emergent ability in terms of breakthrough performance on ERR for the two language pairs under consideration, under a common definition of emergent abilities (Wei et al., 2022a; Schaeffer et al., 2023).

## 6   Related Work

**Automatic Post-Editing of Translations:** There exists a long line of prior work on building neural models for the automatic post-editing (APE) task (Vu and Haffari, 2018; Shterionov et al., 2020; Chatterjee, 2019; Góis et al., 2020; Correia and Martins, 2019a; Voita et al., 2019; Chollampatt et al., 2020; do Carmo et al., 2021). Shterionov et al. (2020) presented a comprehensive road-map for APE, highlighting the challenges and potential directions for future research. Chatterjee (2019) explored the use of deep learning techniques for APE and proposed novel architectures to improve the quality of post-edited translations, while Góis et al. (2020) focused on learning strategies for APE and investigated the use of automatic orderings techniques to refine translations. Correia and Martins (2019b) proposed a simple yet effective neural model for APE using transfer learning, demonstrating promising results.

Voita et al. (2019) introduced a context-aware approach to APE, incorporating source context in-formation into the neural model to generate more accurate post-edits. Chollampatt et al. (2020) explored the use of APE to improve the overall translation quality for NMT models. They investigate the effects of varying training data sizes, using artificial training data, and domain specificity for the APE task. In a comprehensive review, do Carmo et al. (2021) provided an overview of various techniques and approaches in the field of APE, covering both traditional and neural-based methods. Overall, a number of prior studies have explored different architectures, learning strategies, and contextual information integration in neural models (non-LLM) to improve the quality of post-edited translations.

**Leveraging LLMs for Post-Editing:** Vidal et al. (2022) explored the use of GPT-3 based post-editing using glossaries, however, to the best of our knowledge, we present the first work that investigates using GPT-4 for automatic post-editing of translations and presents a formalization of the direct post-editing task under a purely generative setting. Our work is also related to a number of works exploring the using of LLMs for translation (Hendy et al., 2023; Gao et al., 2023; Lu et al., 2023; Vilar et al., 2022; Garcia et al., 2023), however the focus of our work, the task of direct automatic post-editing, is different from existing works.

## 7   Summary and Conclusions

We formalized the task of direct automatic post-editing in a generative setting and posited a set of research questions and measurements to quantify the utility of the state-of-the-art LLM, GPT-4 on the task. Through this formalization, we demonstrated that zero-shot chain-of-thought is critical in constraining the post-edited translation to be close to the initial translation. We also demonstrated that GPT-4 produces meaningful human-judgement aligned edits to translations that also lead to general quality improvements, as evaluated on the WMT-22 test sets. Further, we demonstrated that the edit generation process in GPT-4 is considerably more trustworthy than a previous generation of LLM. Overall, we demonstrated promising results on post-editing with GPT-4, improving upon the WMT-22 Best translation performance on English-Chinese, English-German, Chinese-English and German-English language pairs.

# 8 Limitations

We proposed a formalization to study direct automatic post-editing with state-of-the-art LLMs and investigated a number of research questions through this formalization. However, we only have API-level access to GPT-4. Even though we conducted our experiments on WMT-22 test sets and system outputs, the curation of which falls outside the cut-off date for GPT-4 training data; due to only black-box access to the model, we cannot rule out the possibility of data contamination, even on the WMT-22 test sets.

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

## A Experiments on WMT-20 and WMT-21

In this section, we report the post-editing performance of GPT-4 (Table 13) and gpt-3.5-turbo (Table 14) for En-De, on WMT-20 and WMT-21 system outputs with Major errors as provided by Freitag et al. (2021).

## B Experiments on More Language Pairs

We report further results with GPT-4 based post-editing under the CoT setting in Table 15.

## C Prompts Used for Post-Editing

Tables 16 and 17 list the prompts used for the baselines. The baseline without CoT only uses Step 2 in the user prompt, and the same system prompt as the one in CoT baseline.

## D Edit Hallucinations by GPT-4

Tables 18, 19, 20, 21 present examples where the human annotators reported only a 'PARTIAL' number of edits realized in the final translation.

| System | Initial-QE | PE-QE | E3S |
|---|---|---|---|
| WMT-20 | | | |
| Tohoku | 80.93 | **82.59** | 53.01 % |
| OPPO | 79.25 | **82.32** | 56.11 % |
| eTranslation | 78.82 | **82.72** | 57.04 % |
| Tencent | 80.03 | **82.29** | 52.84 % |
| Huoshan | 78.49 | **82.26** | 56.33 % |
| WMT-21 | | | |
| VolcTrans-GLAT | 80.22 | **82.68** | 39.13 % |
| Facebook-AI | 82.88 | **82.73** | 38.16 % |
| HuaweiTSC | 80.98 | **82.70** | 64.65 % |
| UEdin | 80.82 | **81.34** | 46.67 % |
| eTranslation | 80.04 | **81.60** | 74.05 % |

Table 13: **Edit Efficacy over Erroneous Spans** with **GPT-4**: Post-Editing with GPT-4 modifies more than half of the erroneous spans on average.

| System | Initial-QE | PE-QE | E3S |
|---|---|---|---|
| WMT-20 | | | |
| Tohoku | 80.93 | **82.50** | 69.86 |
| OPPO | 79.25 | **81.59** | 73.49 |
| eTranslation | 78.82 | **82.03** | 73.45 |
| Tencent | 80.03 | **82.49** | 71.48 |
| Huoshan | 78.49 | **82.34** | 71.72 |
| WMT-21 | | | |
| VolcTrans-GLAT | 80.22 | **82.60** | 56.52 |
| Facebook-AI | 82.88 | **82.45** | 60.53 |
| HuaweiTSC | 80.98 | **82.64** | 70.71 |
| UEdin | 80.82 | **81.99** | 74.17 |
| eTranslation | 80.04 | **81.60** | 74.05 |

Table 14: **Edit Efficacy over Erroneous Spans** with **gpt-3.5-turbo**: On both WMT-20 and WMT-21 Systems, post-editing with gpt-3.5-turbo modifies more than half of the erroneous spans.

| Language | System | COMET-KIWI | COMET-QE | COMET-22 | COMET-20 |
|---|---|---|---|---|---|
| En-Ha | MS Translator | 57.18 | 2.68 | 72.51 | -13.04 |
| En-Ha | **MS Translator + GPT-4-CoT** | **59.02** | **3.03** | **74.10** | **-4.27** |
| Ha-En | MS Translator | 68.45 | 15.62 | 73.26 | 13.23 |
| Ha-En | **MS Translator + GPT-4-CoT** | **73.19** | **22.05** | **77.83** | **32.76** |
| En-Ja | MS Translator | 85.27 | 36.80 | 87.95 | 57.97 |
| En-Ja | **MS Translator + GPT-4-CoT** | **85.39** | **38.73** | **89.04** | **62.41** |
| Ja-En | MS Translator | 80.08 | 22.20 | 81.53 | 36.14 |
| Ja-En | **MS Translator + GPT-4-CoT** | **80.92** | **25.00** | **82.93** | **42.96** |
| Cs-En | MS Translator | 82.18 | 32.90 | 87.44 | 72.02 |
| Cs-En | **MS Translator + GPT-4-CoT** | **82.65** | **34.76** | **87.56** | **72.28** |
| En-Cs | MS Translator | 84.16 | **59.02** | **90.63** | **94.06** |
| En-Cs | **MS Translator + GPT-4-CoT** | **84.27** | 58.95 | 90.62 | 93.02 |
| En-Ru | MS Translator | 82.85 | 45.97 | 87.44 | 67.37 |
| En-Ru | **MS Translator + GPT-4-CoT** | **82.94** | **47.75** | **88.05** | **68.56** |
| Ru-En | MS Translator | 80.72 | 31.11 | 85.16 | 62.24 |
| Ru-En | **MS Translator + GPT-4-CoT** | **81.28** | **32.79** | **85.66** | **63.94** |
| En-Is | MS Translator | **80.20** | **34.50** | **84.25** | **65.75** |
| En-Is | **MS Translator + GPT-4-CoT** | 79.27 | 31.96 | 83.47 | 62.45 |
| Is-En | MS Translator | 80.33 | **31.96** | **85.91** | **65.98** |
| Is-En | **MS Translator + GPT-4-CoT** | **81.11** | 31.96 | 83.47 | 62.45 |
| En-Uk | MS Translator | 81.85 | 47.73 | 86.13 | 61.09 |
| En-Uk | **MS Translator + GPT-4-CoT** | **81.96** | **48.89** | **86.96** | **63.08** |
| Uk-En | MS Translator | 79.72 | 25.28 | 83.47 | 52.37 |
| Uk-En | **MS Translator + GPT-4-CoT** | **81.18** | **28.38** | **85.42** | **60.27** |

Table 15: **General Quality Improvements on WMT-22 Test Sets:** The + sign reflects that the post-editing is applied on the initial translations produced by the given System. The post-editing is applied in the CoT setting throughout the results in this table.

---

**System Prompt**

---

You are a native speaker of both English and German. You are an expert post editor of translations from English into German.

You know that the German translation of a given English text must faithfully represent its meaning in German. The English input text itself might contain any number of different words, including typos and placeholder entities, but still the German translation must remain faithful to the English input text. Faithfulness of a German translation means that every word in the translation can be reconstructed from the given English input and vice versa. Therefore, you notice any deviations in the faithfulness of the German translations, including the below issues that make the given German translation not optimal:

1. words in the German translation that are not supported in the input 2. words in the English input that are not adequately translated 3. words in the German translation that do not convey the specific meaning of the corresponding word in the input 4. words in the German translation that are not in the correct language 5. punctuations in the German translation that are different from the input 6. symbols in the English input that are not correctly present in the German translation 7. casing in the German translation that does not conform to the English input 8. incorrect modifications in the German translations of names, organizations, entities 9. incorrect modifications in the German translations of any cardinal or ordinal numbers 10. incorrect translations of web terminologies such as urls, web addresses and hashtags in the input 11. incorrect translations of physical units or currencies in the input 12. unsupported expansions of the acronyms present in the input

You identify and fix the above twelve issues one by one in the German translation if they are present, in a way that improves the translation fluency.

Further, as an expert translation post editor, for the improvements made to the German translation, you make sure that the following principles are followed:

1. No corrections are made that add any word or phrase in the translation which are unsupported in the input 2. The capitalizations in the translation strictly follow the input capitalizations, e.g., acronym capitalizations should not be changed 3. The translation contains the appropriate articles and determiners to follow the specifics in the input 4. Do not leave any symbol, word or phrase in the input text untranslated in the final, improved translation 5. Do not add any extraneous words, phrases, clauses or sentences in the translation that is not supported by the input 6. If the input starts with a non capitalized word, the translation starts with a non capitalized word 7. In the case that the translation is severely inadequate, you generate an improved translation from scratch 8. No end punctuations or full stops are added if such punctuations or full stops are not in the input 9. Do not assume that an acronym is a typo, always err on the side of assuming that the presented input words are not typos 10. Do not replace any entities or placeholders in the translation with fictitious (unsupported) entities 11. If the input contains offensive or lewd words, you still translate them faithfully 12. If the translation misses to convey the meaning of a large part of the input sentence, you include the translation for the missing part

---

**User Prompt**

---

As an expert translation post editor, your task is to improve the German translation for the below English text:

**English**: They were addressed to her son, who has autism and lives in a private care facility, she said. But instead of her son's name inside when you opened them, the letters said Dear Maine's Department of Health and Human Services – in Cincinnati, she told local media.

**German**: Sie waren an ihren Sohn gerichtet, der Autismus hat und in einer privaten Pflegeeinrichtung lebt, sagte sie. Aber anstelle des Namens ihres Sohnes im Inneren, als Sie sie öffneten, hieß es in den Briefen Dear Maine Dear Maine 's Department of Health and Human ServicesServices - in Cincinnati, sagte sie den lokalen Medien.

To accomplish this, follow these steps:

Step 1: Say "Proposed Improvements:". Then brainstorm and design the improvements that make the German translation more faithful and fluent. Step 2: Say "Improved Translation:". Then output the German translation with proposed improvements that increase translation faithfulness and fluency.

---

Table 16: **Post-Editing Prompt**: System and User Prompts for **Post-Editing with Chain-of-Thought** Baseline. The newlines are suppressed in the table.

| **System Prompt** |
|---|

You will work as a machine translation annotator to help assess the quality of translation:

Please identify all errors within each translated sentence, up to a maximum of five. If there are more than five errors, identify only the five most severe. To identify an error, specify the relevant span of text, and select a category/sub-category and severity level from the available options. (The span of text may be in the source sentence if the error is a source error or an omission.) When identifying errors, please be as fine-grained as possible. For example, if a sentence contains two words that are each mistranslated, two separate mistranslation errors should be recorded. If a single stretch of text contains multiple errors, you only need to indicate the one that is most severe. If all have the same severity, choose the first matching category listed in the error typology (eg, Accuracy, then Fluency, then Terminology, etc). Be very precise and accurate.

If there is an error in translation, identify the severity of the error as follows:

Major: Errors that may confuse or mislead the reader due to significant change in meaning or because they appear in a visible or important part of the content. Minor: Errors that don't lead to loss of meaning and wouldn't confuse or mislead the reader but would be noticed, would decrease stylistic quality, fluency or clarity, or would make the content less appealing. Neutral: Use to log additional information, problems or changes to be made that don't count as errors, e.g., they reflect a reviewer's choice or preferred style.

If there is an error in translation, try to place it in a category below. If it doesn't match any of those categories, place it as an Other error:

1. Accuracy: there is an error with the translation accuracy, if it matches any of the following categories: Accuracy/Addition: Translation includes information not present in the source. Accuracy/Omission: Translation is missing content from the source. Accuracy/Mistranslation: Translation does not accurately represent the source. Accuracy/Untranslated text: Source text has been left untranslated.

2. Fluency: there is an error with the translation fluency, if it matches any of the following categories: Fluency/Punctuation: Incorrect punctuation (for locale or style). Fluency/Spelling: Incorrect spelling or capitalization. Fluency/Grammar: Problems with grammar, other than orthography Fluency/Register: Wrong grammatical register (e.g., inappropriately informal pronouns). Fluency/Inconsistency: Internal inconsistency. Fluency/Character encoding: Characters are garbled due to incorrect encoding.

3. Terminology: Terminology is inappropriate or inconsistent: Terminology/Inappropriate: Terminology is non-standard or does not fit context. Terminology/Inconsistent: Terminology is used inconsistently.

4. Style: Translation is awkward with stylistic problems.

5. Locale convention: Wrong format for addresses, currency, dates, names, telephone numbers or time expressions. Locale/Address: Wrong format for addresses. Locale/Currency: Wrong format for currency. Locale/Date: Wrong format for dates. Locale/Name: Wrong format for names. Locale/Telephone: Wrong format for telephone numbers. Locale/Time: Wrong format for time expressions.

After identifying all the errors, you will produce an improved translation that fixes the identified errors. For the improvements made to the translation, you make sure that the following principles are followed: 1. No corrections are made that add any word or phrase in the translation which are unsupported in the input 2. The capitalizations in the translation strictly follow the input capitalizations, e.g., acronym capitalizations should not be changed 3. The translation contains the appropriate articles and determiners to follow the specifics in the input 4. Do not leave any symbol, word or phrase in the input text untranslated in the final, improved translation 5. Do not add any extraneous words, phrases, clauses or sentences in the translation that is not supported by the input 6. If the input starts with a non capitalized word, the translation starts with a non capitalized word 7. In the case that the translation is severely inadequate, you generate an improved translation from scratch 8. No end punctuations or full stops are added if such punctuations or full stops are not in the input 9. Do not assume that an acronym is a typo, always err on the side of assuming that the presented input words are not typos 10. Do not replace any entities or placeholders in the translation with fictitious (unsupported) entities 11. If the input contains offensive or lewd words, you still translate them faithfully 12. If the translation misses to convey the meaning of a large part of the input sentence, you include the translation for the missing part

Table 17: **Post-Editing Prompt**: System and User Prompts for Post-Editing with **Structured Chain-of-Thought** Baseline. The user prompt remains the same throughout. The newlines are suppressed in the table.

**Example of Post-Editing using GPT-4 (Verbatim Outputs under the CoT Setting)**

**Source**: Wir wollten jene mit Tischplatten Durchmesser von #NUMBER#
**Translation**: We wanted the ones with table tops diameter of #NUMBER#

**Proposed Improvements**:
1. Change "table tops diameter" to "tabletop diameters"
2. Add "a" before "diameter"
**Improved Translation**:
We wanted the ones with tabletop diameters of #NUMBER#

Table 18: **Edit Hallucination Example on a WMT-22 De-En Sample**: GPT-4 produces an unrealized edit.

---

**Example of Post-Editing using GPT-4 (Verbatim Outputs under the CoT Setting)**

**Source**: Der Händler teilt dem Kunden die Kosten des Teils vor der Reparatur mit.
**Translation**:Der Händler teilt dem Kunden die Kosten des Teils vor der Reparatur mit.

**Proposed Improvements**:
1. Change "will notify" to "informs" to better convey the present tense of "teilt...mit" in the German input.
2. Add "with" after "informs" to better convey the meaning of "teilt...mit" in the German input.
**Improved Translation**:
The dealer informs the customer of the cost of the part before the repair.

Table 19: **Edit Hallucination Example on a WMT-22 En-De Sample**: GPT-4 produces an unrealized edit.

---

**Example of Post-Editing using GPT-4 (Verbatim Outputs under the CoT Setting)**

**Source**: schicken Sie mir bitte irgendeine Transaktionsnummer.. oder sonst was, was dabei helfen könnte
**Translation**: please send me any transaction number.. or anything else that could help

**Proposed Improvements**:
1. Replace the double period with a single period after "transaction number.
2. Add a comma after "or anything else".
**Improved Translation**:
Please send me any transaction number or anything else that could help.

Table 20: **Edit Hallucination Example on a WMT-22 En-De Sample**: GPT-4 produces an unrealized edit.

---

**Example of Post-Editing using GPT-4 (Verbatim Outputs under the CoT Setting)**

**Source**: Der Spiegel hat einen Kratzer und am Rahmen ist ebenfalls ein Fehler, der mit einem gelben Füllstoff versucht wurde kaschiert zu werden.
**Translation**: The mirror has a scratch and there is also an error on the frame that was attempted to be covered with a yellow filler.

**Proposed Improvements**:
1. Replace "error" with "blemish" to better convey the meaning of "Fehler" in this context.
2. Rearrange the sentence structure to improve fluency.
**Improved Translation**:
The mirror has a scratch, and there is also a blemish on the frame that was attempted to be concealed with a yellow filler.

Table 21: **Edit Hallucination Example on a WMT-22 En-De Sample**: GPT-4 produces an unrealized edit.