# OpenReview forum: "Leveraging GPT-4 for Automatic Translation Post-Editing"
_EMNLP/2023/Conference — EMNLP 2023 Findings_

### Official Review · Reviewer_NRkU · 2023-07-28

**Soundness:** 4

**Excitement:**

4: Strong: This paper deepens the understanding of some phenomenon or lowers the barriers to an existing research direction.

**Paper Topic And Main Contributions:**

The paper explores using a large language model, specifically GPT-3.5 and GPT-4, for automatic translation post-editing across multiple language pairs. This task involves refining and improving translations produced by Neural Machine Translation (NMT) Models.

The main contributions of the paper are:
* The paper demonstrates the efficacy of using GPT-4 for post-editing by showing state-of-the-art performance on several language pairs using state-of-the-art Machine Translation quality metrics.
* The paper also introduces several measurements to evaluate the efficacy of post-editing. These include Translation Edit Rate (TER) to measure the nature of the post-edited translation, various standard Translation quality metrics (COMET) to measure general quality improvements, Edit Efficacy over Error Spans (E3S) to measure edits on human-annotated error spans, and Edit Realization Rate (ERR) to calculate the fidelity of the proposed edits.
* The paper discusses the benefits of using LLMs for post-editing, like advanced multi-lingual understanding capabilities, and their potential for knowledge-based or culture-specific customization.

**Reasons To Accept:**

* The authors showcase a significant quality improvement of their post-editing over the NMT translation across multiple languages, which is substantial since the LLM translations are worse than the NMT translations.
* The investigation of the chain of thoughts for machine translation quality and the fidelity of the proposed edits is very insightful. It questions the zero-shot chain of thought step for the post-editing task.
* The paper provides a thorough literature review and related work, demonstrating a profound understanding of the field.

**Reasons To Reject:**

* The paper could have provided a clearer use case for why the NMT model is still necessary. When powerful and large general language models are used for post-editing, why should one use a specialized neural machine translation model to get the translations in the first place? Including GPT-4 translations plus GPT-4 post-editing scores would have been insightful.
* The paper lacks an ablation study explaining why they chose the prompt in this specific way, e.g., few-shot examples for CoT might improve performance.
* The reliance on an external model via API, where it's unclear how the underlying model changes, makes it hard to reproduce the results. There is also a risk of data pollution since translations might already be in the training data of GPT-4. The authors only state that the WMT-22 test data is after the cutoff date of the GPT-4 training data, but they do not say anything about the WMT-20 and WMT-21 datasets that they also use.
* The nature of post-edited translation experiments is only partially done: En-Zh and Zh-En for GPT-4 but not for En-De and De-En for GPT-3.5.
* The title is misleading since the authors also evaluate GPT-3.5.

**Reproducibility:**

2: Would be hard pressed to reproduce the results. The contribution depends on data that are simply not available outside the author's institution or consortium; not enough details are provided.

**Reviewer Confidence:**

5: Positive that my evaluation is correct. I read the paper very carefully and I am very familiar with related work.

---

> ### Author Rebuttal · Authors · 2023-08-29
>
> We thank the reviewer for the detailed comments and appreciation of our paper. Please find our response below:
>
> 1. Use of non-LLM Translations: While state of the art LLMs such as GPT-4 could be used to obtain translations, NMT systems are computationally far cheaper to run at scale and could be run on CPUs as well. Thereby, obtaining translations directly through GPT-4 is practically less beneficial than its use as post-editor of translations in critical usage settings. In the camera-ready revision, we will include experiments on GPT-4 based post-editing on GPT-4 translations in the appendix.
>
> 2. Ablation on Prompts: We report results on zero-shot post-editing since few-shot requires additional manual labor in crafting the demonstrations (besides just crafting the prompt as in zero-shot) on a per language pair basis, hence is not as scalable as zero-shot post-editing. This is besides the fact that acquiring high-quality post-editing demonstration data on a per language pair basis is difficult, especially in the CoT setting. Hence, we do not report few-shot results and leave few-shot data selection/creation for the formalized post-editing task to future work.
>
> 3. WMT-20, 21 Experiments and Reproducibility: None of our claims rely exclusively on WMT-20 or WMT-21 test data, since the vast majority of our experiments are conducted on WMT-22. Further, we have included the prompts used in the appendix and we will make the prompt files public upon acceptance. We acknowledge that using a pure white-box LLM would have been better, but at this point state-of-the-art LLMs (especially on multilingual tasks) are not open source and owing to the widespread use of OpenAI APIs, we believe that our results could be readily reproduced.
>
> 4. Nature of Post-Editing Experiments: Thanks for pointing this out. We had conducted experiments on En-De and De-En as well as with gpt-3.5-turbo over the post-edited outputs of more than 10 systems but omitted it from the main draft. We will include these results in the appendix. Throughout the experiments, we found that CoT constrains the post-edited translations to be closer (as measured through TER) to the initial translations, and this holds true across the language pairs and for both gpt-4 and gpt-3.5-turbo. We will also include this measurement for all the WMT-22 language pairs (18) experimented on, in the revision.
>
> 5. Title: Thanks for the suggestion, we agree that we have conducted a much more comprehensive study than the title suggests, and we will modify the title to reflect this.

---

### Official Review · Reviewer_Cpb4 · 2023-08-01

**Soundness:** 3

**Excitement:**

4: Strong: This paper deepens the understanding of some phenomenon or lowers the barriers to an existing research direction.

**Missing References:**

none

**Paper Topic And Main Contributions:**

The authors formalize the task of direct translation post-editing with LLMs and explore the use of GPT-4 to automatically post-edit NMT outputs across several language pairs. They find the GPT-4 is superior to other LLMs but could prodice hallucinated edits which urges caution.

* The authors formalize the post-editing task in a generative setting with LLM, withtout intermediate error detection steps.
* They conduct comprehensive experiments and analyses under the new setting, and find that GPT4-based method can achieve better performance.
* This might be the first work investigating GPT-4 etc. LLMs on post-editing tasks which provides insights for future research.


**Questions For The Authors:**

* Is the CoT content (proposed improvements in Table 1 and 12), i.e., the "E" in line 87, created by LLMs? This part is not that clear, do you feed "proposed improments:" and "improved translation:" to LLMs?
* have you tried using LLMs with demonstrations/examples instead of zero-shot setting?
* RQ1, lines 115-119: I do not agree with this assumption, because the post-edited translations may be follows the revision suggestions but close to the translations produced by LLMs directly.

**Reasons To Accept:**

* The authors formalize the post-editing task in a generative setting with LLM, withtout intermediate error detection steps. This might be the first work investigating GPT-4 etc. LLMs on post-editing tasks which provides insights for future research.
* They conduct comprehensive experiments and analyses under the new setting, and find that GPT4-based method can achieve better performance.


**Reasons To Reject:**

* some parts are not clearly described, such as the CoT settings are not clear to me (detailed questions are in the questions section).
* the utilization of LLMs seems too simple, some more advanced prompts should be further investigated.

**Reproducibility:**

4: Could mostly reproduce the results, but there may be some variation because of sample variance or minor variations in their interpretation of the protocol or method.

**Reviewer Confidence:**

4: Quite sure. I tried to check the important points carefully. It's unlikely, though conceivable, that I missed something that should affect my ratings.

**Typos Grammar Style And Presentation Improvements:**

* providing average scores for each system in table 5-8 would be better for readers.
* give some examples to illustrate the Structured-CoT would be more friendly to readers and it is crucial to your experiments.

---

> ### Author Rebuttal · Authors · 2023-08-29
>
> We thank the reviewer for the detailed comments. Please find our response below:
>
> 1. CoT Setting: Yes, the ‘E’ in line 87 is created by LLMs. Table 1 and 12 list the verbatim outputs produced by the LLM, i.e., the entire string after the source and translation entry is produced by the LLM as single-step generation. The prompt used for the CoT setting is highlighted in Table 16 in Appendix (Page 14). We do not feed “Proposed Improvements” and “Improved Translation” as separate steps but prompt the LLM to produce its output in this particular format by feeding this format directly in the prompt.
>
> 2. Example on Structured-CoT: Table 17 lists the prompt used for structured-CoT, which prompts LLM to produce the improvements in the form of MQM annotations. Here is an example for German-English post-editing under the structured-CoT setting:
>
> Source: Halten Sie sich unbedingt an die angegebenen Erwärmungshinweise, da längere Zeiten oder höhere Energieleistungen zu Schäden führen können (Entflammungs- und Verbrennungsgefahr!).
> Translation: Be sure to follow the heating instructions given, as longer periods or higher energy outputs can lead to damage (risk of ignition and burns!).
> Proposed Improvements:
> 1. Error Category: Accuracy/Mistranslation, Severity: Minor, Proposed Improvement: Change "heating instructions" to "specified warming instructions".
> 2. Error Category: Accuracy/Mistranslation, Severity: Minor, Proposed Improvement: Change "risk of ignition and burns!" to "risk of ignition and combustion!").
> Improved Translation: Be sure to follow the specified warming instructions given, as longer periods or higher energy outputs can lead to damage (risk of ignition and combustion!).
>
> Please note that here, the proposed improvements are generated in the form of MQM annotations, with error categories and severities. Lines 361-375 present a discussion of structured-CoT performance.
>
> 3. RQ1, lines 115-119: We do not pose it as an assumption, we posit it as a question that has to be quantified.
>
> 4. Few-Shot Demonstrations: We report results on zero-shot post-editing since few-shot requires additional manual labor in crafting the demonstrations (besides just crafting the prompt as in zero-shot) on a per language pair basis, hence is not as scalable as zero-shot post-editing. This is besides the fact that acquiring high-quality post-editing demonstration data on a per language pair basis is difficult, especially in the CoT setting. Hence, we do not report few-shot results and leave few-shot data selection/creation for the formalized post-editing task to future work. It is conceivable that high-quality few-shot demonstrations will improve the post-edited translation quality, but it is an empirical question for which we do not have high-quality data under the different settings/language-pairs. On the other hand, the simplicity and effectiveness of zero-shot, as demonstrated is much more scalable.

---

### Official Review · Reviewer_aKRm · 2023-08-07

**Soundness:** 3

**Excitement:**

2: Mediocre: This paper makes marginal contributions (vs non-contemporaneous work), so I would rather not see it in the conference.

**Paper Topic And Main Contributions:**

The authors evaluate the use of the commercial LLM GPT-4 for automatic post-editing (APE). The use of instruction fine-tuned LLMs for the APE task is appealing because of the recent popularity and general excellent performance of GPT-4 in particular on a wide array of zero-shot settings for NLP tasks, including APE. The authors evaluate GPT-4s APE performance on a broad suite of post-editing tasks from WMT.

**Questions For The Authors:**

how did you arrive at the prompt template in appendix C?

**Reasons To Accept:**

- Thorough evaluation methodology including human evaluation
- Very relevant topic to EMNLP participants given the recent popularity and large advancements of general purpose LLMs


**Reasons To Reject:**

- Low scientific contribution / novelty
- Reliance upon commercial, closed-source models, that are not free
- Lack of exploration of related topics of interest: fine-tuning open LLMs for APE, prompt engineering, multilingual LLMs, etc…


**Reproducibility:**

2: Would be hard pressed to reproduce the results. The contribution depends on data that are simply not available outside the author's institution or consortium; not enough details are provided.

**Reviewer Confidence:**

4: Quite sure. I tried to check the important points carefully. It's unlikely, though conceivable, that I missed something that should affect my ratings.

---

> ### Author Rebuttal · Authors · 2023-08-29
>
> We thank the reviewer for the feedback. However, we strongly disagree with the reviewer's characterization of the scientific merit of our work, which stands in contrast to the agreement among reviewers Cpb4 and NRkU who have appreciated our systematic formalization of the post-editing problem in the generative setting and have found our experiments very insightful. We address the points raised below:
>
> 1. Scientific Contribution: We have formalized the post-editing problem in a generative setting and quantified both the utility and shortcomings of existing state of the art LLMs on this problem. Our experiments have yielded quantifiable insights on the utility of zero shot chain of thought as well as on the nature of the post-edited translations, besides producing state of the art performance on WMT-22 benchmark across multiple language pairs.
>
> 2. Closed Source Models: In the space of state-of-the-art LLMs, open-source models lag behind in performance, especially on unseen and multilingual tasks (Holistic Evaluation of Language Models, Liang et al). We believe that getting results on models that demonstrate considerably inferior quality is a disservice to our scientific study of the post editing problem using state-of-the-art LLMs. We acknowledge the fact that open-source models not being at the same capability levels as closed-source models is a cause of concern for the NLP community, but we strongly defend our choice of LLMs for the task under study.
>
> 3. Related Topics of Interest: We indeed have crafted specific prompts for the task of automatic post-editing (prompt-engineering is a starting point of leveraging LLMs) and use the state-of-the-art multilingual LLM for our experiments, but the other suggested points are not very relevant to the research questions we investigate.
>
> 4. Prompt Template in Appendix C: We arrived at the prompt template by positing the specifications of "correct" translations, using which the LLM can reason about the translation. The idea of leveraging specifications for judging the correctness of translations has been in the literature for a long time (e.g., Quality Estimation, Specia et. al, SALTED, Raunak et al.) and we were inspired by existing descriptions of "correct" translations when writing the prompts for post-editing. We will include a section in the appendix to highlight the etymology of each of the components used in our prompt.

---

### Meta-Review · Area_Chair_9d7g · 2023-09-20

**Recommendation:** 4

**Metareview:**

The authors formalize the task of direct translation post-editing with LLMs and explore the use of GPT-4 to automatically post-edit NMT outputs across several language pairs. The authors formalize the post-editing task in a generative setting with LLM, without intermediate error detection steps. They demonstrate that they can improve translations generated with Microsoft Translator and the WMT22 submissions via post-editing across several language pairs. This is the first work investigating a LLM as a post-editing system for MT. The author did a fantastic job with evaluation including a human evaluation.
The main criticism of this work is the reliance on an external model via an API, where it's unclear how the underlying model changes, making it hard to reproduce the results. It would have been nice to also include experiments with open source LLMs and even try to improve their post-editing quality via e.g. fine-tuning on post-edited data.

---

### Decision · Program_Chairs · 2023-10-07

**Decision:**

Accept-Findings

**Comment:**

The authors formalize the task of direct translation post-editing with LLMs and explore the use of GPT-4 to automatically post-edit NMT outputs across several language pairs. The authors formalize the post-editing task in a generative setting with LLM, without intermediate error detection steps. They demonstrate that they can improve translations generated with Microsoft Translator and the WMT22 submissions via post-editing across several language pairs. This is the first work investigating a LLM as a post-editing system for MT. The author did a fantastic job with evaluation including a human evaluation.
The main criticism of this work is the reliance on an external model via an API, where it's unclear how the underlying model changes, making it hard to reproduce the results. It would have been nice to also include experiments with open source LLMs and even try to improve their post-editing quality via e.g. fine-tuning on post-edited data.